# Sustainable Use of Waste Oyster Shell Powders in a Ternary Supplementary Cementitious Material System for Green Concrete

**DOI:** 10.3390/ma15144886

**Published:** 2022-07-13

**Authors:** Shanglai Liu, Yannian Zhang, Bonan Liu, Zhen Zou, Qiang Liu, Yina Teng, Lei V. Zhang

**Affiliations:** 1College of Fisheries and Life Science, Dalian Ocean University, Dalian 116086, China; lsl19904019001@163.com; 2School of Civil Engineering, Shenyang Jianzhu University, Shenyang 110168, China; z15898274285@163.com (Y.Z.); m18342831904@163.com (B.L.); zh1377746456@163.com (Z.Z.); l455535123@outlook.com (Q.L.); 3Changxin International Art School, Yunnan University, Kunming 650106, China; t15142587827@163.com; 4School of Civil and Transportation Engineering, Hebei University of Technology, 5340 Xiping Road, Tianjin 300401, China; 5Department of Civil and Environmental Engineering, Western University, Richmond St., London, ON N6A 3K7, Canada

**Keywords:** oyster shell powder, compressive strength, permeability, coupling effect, supplementary cementitious materials

## Abstract

The increasing concern for decarbonization and sustainability in construction materials is calling for green binders to partially replace cement since its production is responsible for approximately 8% of global anthropogenic greenhouse gas emissions. Supplementary cementitious materials (SCMs), including fly ash, slag, silica fume, etc., can be used as a partial replacement for ordinary Portland cement (OPC) owing to reduced carbon dioxide emissions associated with OPC production. This study aims to investigate the sustainable use of waste oyster shell powder (OSP)-lithium slag (LS)-ground granulated blast furnace slag (GGBFS) ternary SCM system in green concrete. The effect of ternary SCMs to OPC ratio (0%, 10%, 20%, and 30%) on compressive strength and permeability of the green concrete were studied. The reaction products of the concrete containing OSP-LS-GGBFS SCM system were characterized by SEM and thermogravimetric analyses. The results obtained from this study revealed that the compressive strength of concrete mixed with ternary SCMs are improved compared with the reference specimens. The OSP-LS-GGBFS ternary SCMs-based mortars exhibited a lower porosity and permeability compared to the control specimens. However, when the substitution rate was 30%, the two parameters showed a decline. In addition, the samples incorporating ternary SCMs had a more refined pore structure and lower permeability than that of specimens adding OSP alone. This work expands the possibility of valorization of OSP for sustainable construction materials.

## 1. Introduction

Carbon neutrality, as a new commitment to environmental protection and sustainability, has been adopted by more and more countries [1]. To limit global warming and achieve the net-zero target by 2050, according to the Paris Agreement, many countries are committed to carbon offsetting and reducing carbon dioxide emissions. At present, OPC and concrete production results in substantial greenhouse gas emissions owing to their large-scale use. The increasing concern for the carbon giants is calling for sustainable construction materials [2]. Some wastes, including construction wastes [3], industrial solid wastes [4,5], agricultural wastes [6], waste glass [7], rubbers [8], etc., have the potential to replace OPC and aggregates in concrete partially, which not only solves the shortage of concrete raw materials but also alleviates the environmental pressure caused by waste accumulation [9,10,11].

The recycling of shellfish wastes [12] generated by the development of marine resources is rare, despite millions of tons of shell wastes being produced and abandoned in landfills in China every year [13]. These landfilled shells leach heavy metals from the heap due to severe land and air pollution emissions during microbial decomposition and weathering. In some African countries, abandoned shells are difficult to handle, and millions of tons of abandoned shells pile up on shores and beaches each year. Open-air accumulation of abandoned shells is a potential habitat for microbes that attract organisms that are the carriers of possible diseases, causing public health problems [14]. The practical solution to this problem is to recycle these shells as raw materials to develop new green building materials [15]. In China, however, the utilization of discarded shells is less than 30%. Some researchers [16] have applied waste shells as concrete aggregate in building materials. Whether used as coarse aggregate or fine aggregate, the crushing index of the burned waste shell is challenging to meet the standard of traditional sandstone, which significantly limits the application of waste shells in the field of concrete. Shells contain a large amount of calcium oxide and calcium carbonate with recycling value [17]. Extensive research for using OSPs in cement products has been undertaken. For example, Soltanza et al. [18] found the OSP can be used to develop mixed cement and improve the performance of concrete and effectively reduce the heat of hydration cement. Ali et al. [19] found that the OSP enhanced the compressive strength at an early age, but excessive addition will decrease compressive strength. It greatly limits the application prospects of OSP. Abdelaziz et al. [20] conducted alkali-activation treatment on OSP and prepared a new type of alkali-activated shell waste (AASW) mortar with a compressive strength of 22 MPa and a porosity of 16.5%. Hassan et al. [21] evaluated the effect of OSPs on chloride penetration of cement materials. OSP partially replacing cement changed the microstructure of materials and allowed more ions to react with chloride through compaction, thereby reducing the corrosion risk of concrete reinforcement. Therefore, OSP has the potential to develop into supplementary cementitious materials (SCMs) [22]. Pusit et al. [23] used OSPs instead of cement to prepare plaster mortar. Compared with OPC, the compressive strength of the OSP-based concrete was lower in the initial stage; however, the shrinkage rate and thermal conductivity were decreased. Although OSPs can be potentially used as SCM, the compressive strength deteriorates significantly with the increase in the substitution rate. Bassam et al. [24,25] investigated the possibility of replacing cement by grinding and burning bivalve clam seashells. Their results showed that the 5% replacement mix is the optimum percentage of replacement. However, these values are increased with higher levels of replacement.

However, the low volcanic ash activity still limits the application of OSP. While OSPs as SCM have been a topic of intensive research, very few studies have addressed the properties of a multi-SCM system containing OSPs [26]. Ground granulated blast furnace slag (GGBFS) and lithium slag (LS) are two types of wastes from iron and lithium industries. Due to the high activity, they are widely used in cement concrete products. Shariq et al. [27] explored the compressive strength at various ages of the concrete specimens containing GGBFS with different amounts. It was found that the GGBFS concrete with 40% replacement of OPC achieved the highest compressive strength at the age of 56 days. He et al. [28] showed that the microstructure of the UHPC specimens with appropriate content of LS was improved at late ages; however, the addition of LS affected negatively at early ages. In addition, the use of LS accelerated the hydration of UHPC and increased elastic modulus of the interfacial transition zone (ITZ) in UHPC.

Although the efficacy of OSPs in concrete has been well-documented and understood, there remains doubts and ongoing controversy regarding whether a multi-SCM system containing OSPs are applicable. As such, systematic research is needed to acquire a better understanding of OSPs in SCM system implemented in concrete and to elucidate the performance, thereby recycling resources and reducing carbon dioxide emissions for sustainable construction. In this study, OSP combined with LS and GGBFS as a ternary SCM system were incorporated into mortars. The present paper attempted to study the use of the OSP-LS-GGBFS ternary system as a partial replacement of cement with different effect substitution ratios by weight. The compressive strength and permeability of the mortars with the OSP-LS-GGBFS ternary system were studied. In addition, the hydration degree, pore structure, and microstructure were also explored. This study provides a new understanding of OSP in developing sustainable construction materials and demonstrates the compatibility of the three types of SCMs in green concrete.

## 2. Materials and Methods

### 2.1. Raw Materials

The raw materials include OSP, GGBFS, LS, Portland cement (PO.42.5), standard sand, and water. PO.42.5 grade cement from Shenyang Shanshui Park Cement Co., Ltd. (Shenyang, China) Was used as a primary binder in this study. Oyster shells were provided by Rongcheng Xingyang Fish Flour Factory in Rongcheng, Shandong Province, China. The shells were cleaned with water to remove any residues [29] that may remain in the shells, then dried at 105 ± 5 °C for 24 h to remove any moisture content, and the OSP samples were then calcined for 2 h at 850–950 °C, as recommended by Li et al. [30] According to Letwattanaruk et al. [31], who used OSP as SCM, the average particle size range should be from 0.5 to 40 μm. GGBFS was sourced from Jiyuan Steel Plant in Jiyuan, Henan Province, China. LS was obtained from Tianyuan New Energy Materials Co., Ltd. located in Taiyuan, Guangxi Province, China. The chemical compositions of the three types of materials are shown in Table 1. The specific surface area in Table 2 and the particle size distribution of OSP, LS, and GGBFS are shown in Figure 1 via Brunauer–Emmett–Teller (BET) surface area analyses. The XRD in OSP was tested ranging from 5° to 60° 2θ. Figure 2 shows that the main mineral composition of the OSP is dolomite, followed by quartz. 

### 2.2. Mix Design and Specimens Preparation

Table 3 and Table 4 present the mix details of the specimens. In the M group, the coupling effect among OSP, GGBFS, and LS was explored, and m-1 was the control group of OSP alone. M-2 is a binary combination of OSP and LS, M-3 is a binary combination of OSP and GGBFS, m-4 is a ternary combination of OSP, LF, and GGBFS. In group D, the influence of SCMs replacement on the asp-GGBFS-LS ternary system was explored. For D-0, the replacement rate of standard group was 0%; for D-10, D-20, and D-30, the replacement rate was 10%, 20%, and 30%. The required raw materials were poured into a JJ-5 planetary mixer to produce mortars as per China standard GB/T17671-1999 [32]. The mortars were placed in molds with a dimension of 40 mm × 40 mm × 160 mm and vibrated for 120 s. All specimens were demoulded after 24 h and then cured in a standard curing room at 20 ± 2 °C with relative humidity greater than 95%. Cubic 40-mm specimens were cast for measuring compressive strength. The purified slurry samples were crushed and soaked in anhydrous ethanol for 3 days to terminate hydration. The purified slurry samples were then dried in a vacuum drying chamber at 50 °C and further characterized.

### 2.3. Experimental Procedures

#### 2.3.1. Compression Tests

Compression tests were performed via GYE-300B universal testing machine (Beijing Kodak Jinwei Technology Development Co., Ltd., Beijing, China). A loading rate of 2.4 kN/s, as stipulated by GB/T17671-1999, was conducted during the loading. 

#### 2.3.2. Thermogravimetric Analysis (TGA)

TGA was conducted using TA instruments (Q500 V20.13, USA) in a nitrogen gas flow with a 70 mL/min flow rate. Each testing sample was about 15 mg. The TGA was exposed to the temperature from room temperature to 800 °C with a rate of 20 °C/min.

#### 2.3.3. Permeability Tests

(1)
*Water absorption*


The test was carried out according to the British standard. The cylinder sample (U75 mm × 75 mm) was solidified and demoulded. The standard curing times were 7 days and 28 days. The sample was dried in an oven (100 °C) for 3 days and cooled for 1 day. The sample was weighed, immersed in water for 24 h, and weighed again after 24 h. The average value of the two samples was the final result of each sample. The hygroscopic rate was calculated according to Equation (1):(1)$$W=\frac{W_{w}-W_{d}}{W_{d}}\times 100\%$$
where *W* = hygroscopic rate; *W_w_* = weight of wet sample; *W_d_* = weight of dry sample.

(2)
*Rapid chloride penetration test (RCPT)*


The chloride ion penetration test was carried out according to ASTM C1202 [33], and the chloride ion charge was determined. The cylinder sample was 100 mm in diameter and 50 mm in length. After 28 days of water curing, the epoxy resin was smeared on the side, and the sample was fixed between two specific grooves. One side of the groove was connected with the sample, and the other side was connected with the solution to connect the groove containing NaCl to the negative electrode, and the groove containing NaOH solution was connected to the positive electrode. The experiment started with DC 60 V, and the total current passed through the sample was measured for 6 h, which was automatically recorded every 5 min. The chloride permeability rating [34] is shown in Table 5.

#### 2.3.4. Mercury Intrusion Porosimetry (MIP)

The pore size distribution of all specimens at the ages of 28 days was investigated using Mercury intrusion porosimetry (MIP) (Micrometrics AutoPore IV 9500, Norcross, GA, USA). The sample was dried in a 60 °C oven and tested three days later. The relationship between the applied pressure and the cylindrical aperture was described by the Washburn equation [35].

#### 2.3.5. Scanning Electron Microscopy (SEM)

The microstructure of the hydration products of the paste samples at 28 days was analyzed via SEM. SEM test was conducted using QUANTA 200 FEG SCANNING electron microscope produced by FEI Company of Holland (Eindhoven, The Netherlands), and the acceleration voltage was 0.2–30 kV. All specimens were immersed in anhydrous ethanol for 3 days before the SEM test to prevent water and reaction, and then the samples were dried in the oven to remove anhydrous ethanol. 

## 3. Results and Discussion

### 3.1. Compressive Strength Analysis

Figure 3 illustrates the coupling effect of a ternary SCM system on the compressive strength considering SCM types (see Figure 3a) and replacement ratios (see Figure 3b) in the mix. As shown in Figure 3a, single incorporation of OSP as SCM (M-1) caused the lowest compressive strength, which implies that OSP alone cannot provide pozzolanic reactions for refining the pore structure, thereby improving the compressive strength. In addition, the addition of calcined OSP increased the alkalinity in the matrix owing to the high content of calcium hydroxide. The increase in alkalinity affects the density of the C-S-H gel and causes a series of durability problems, thereby deteriorating the microstructure. The results from the study by Jian et al. [36] showed that pore solutions with different alkalinity affected the morphology of C-S-H, thus influencing the microstructure. It was found that the ternary system (M-4) resulted in the highest compressive strength, which was 26% higher than that of M-1 at 28 days. In contrast, the compressive strengths of the specimens from GGBFS and LS-based binary systems (M-2 and M-3) were lower than that of M-4. This implies that a ternary SCM system played a synergistic role in refining the microstructure of the matrix, thereby improving the compressive strength. As shown in Table 1, OSP has the highest content of CaO, and LS has a large proportion of Al_2_O_3_. SiO_2_ is the main oxide in GGBFS. LS and GGBFS both have volcanic ash properties. When CaO in calcined OSP becomes CH, it further reacts with Al_2_O_3_ in LS and SiO_2_ in GGBFS for the coming secondary hydration [37,38], according to Equations (2) and (3).
(2)$$x\;Ca{\left(OH\right)}_{2}+SiO_{2}+mH_{2}O\to xCaO\cdot SiO_{2}\cdot mH_{2}O$$
(3)$$x\;Ca{\left(OH\right)}_{2}+Al_{2}O_{3}+nH_{2}O\to xCaO\cdot Al_{2}O_{3}\cdot mH_{2}O$$

As can be seen in Figure 3b, compared with the reference group, the compressive strength of the specimens incorporating 10% of cement was slightly improved. For example, the compressive strength of D-0 at the age of 14 days increased by 10%. When further increasing the replacement ratio, there was a downward trend in the compressive strength. A slight improvement in the strength development incorporating the ternary SCMs with a 10% replacement ratio was likely due to the pozzolanic reactions and micro-aggregate filling effect [39]. The declining trend in compressive strength when replacing large amounts of cement was probably attributed to the low binding capability of SCMs in the initial stage given the low content of cement. A similar result was reported by Muthusamy et al. [40], who found that the compressive strength decreased with the increasing replacement rate of OSPs.

### 3.2. Hydration Products Analysis

#### 3.2.1. TGA

During the TG test of hardened cement-based materials, different types of hydration products will dehydrate or decompose at different temperatures [41]. The binding water of the C-S-H gel and the Aft particles began to dehydrate at 60–300 °C and CH began to dehydrate at 350–550 °C. CaCO_3_ in the slurry with carbonation begins to decompose at a temperature of 600–800 °C [42]. Therefore, the content of corresponding substances was quantitatively calculated by measuring the mass loss at a specific temperature stage.

Figure 4 shows the TG results of each experimental group at 28 days. In each TG test, the first endothermic peak occurs between 100–120 °C. This can be attributed to dehydration of the C-S-H gel. It is worth mentioning that in the TG image of the ternary system, there is no obvious endothermic peak formed by Aft dehydration near 200 °C, and only the slope changes, indicating that the Aft content in the ternary system is less than that in the late hydration of pure cement paste. At 460–480 °C, there is a second endothermic peak. This is caused by the dehydration of CH by heat. At 690–730 °C, there is a third endothermic peak. This is due to the decomposition of CaCO_3_ formed by the carbonization of CH [43].

In order to obtain the hydration process and hydration product content of the ternary system clearly, the C-S-H and CH content of each group at 28 days were calculated by Equations (4) and (5) [44].
(4)$$\mathrm{CH}={\mathrm{WL}}_{CH}\times \frac{m_{CH}}{m_{H_{2}O}}+WL_{CaCO_{3}}\times \frac{m_{CaCO_{3}}}{m_{CO_{2}}}$$
(5)$$H_{2}O={\mathrm{WL}}_{CSH}+{\mathrm{WL}}_{CH}+WL_{CaCO_{3}}\times \frac{m_{H2O}}{m_{CO_{2}}}\times \frac{2}{3}$$
where CH denotes the sample’s relative calcium hydroxide content; H_2_O is the sample’s relative water content; The mass loss of calcium carbonate caused by the removal of water by TG is denoted by WL_CH_, %; WL_CaCO3_ denotes the mass loss of calcium carbonate due to water removal by TG; The mass loss of calcium carbonate caused by the removal of water by TG is denoted by WL_C-S-H_; The molar mass of calcium hydroxide is denoted by m_CH_; The molar mass of water is denoted by m_H2O_; m_CaCO3_ is the molar mass of calcium carbonate; m_CO2_ is the molar mass of carbon dioxide.

Substitute the data from TG images of the ternary system into Table 6 using Equations (6) and (7) as shown below:(6)$$\mathrm{CH}={\mathrm{WL}}_{CH}\times \frac{74}{18}+WL_{CaCO_{3}}\times \frac{100}{44}$$
(7)$$H_{2}O={\mathrm{WL}}_{CSH}+{\mathrm{WL}}_{CH}+WL_{CaCO_{3}}\times \frac{6}{22}$$

Table 6 shows the H_2_O and CH content in each experimental group. Gu et al. [45] found the content of chemically bonded water in cement-based cementitious materials can reflect the degree of hydration in the system, and the higher the degree of hydration, the more chemically bonded water content. The chemically bound water in the TG tests showed an irregular distribution because of the distribution of chemically bound water in C-S-H, CH, and CaCO_3_. Part of CH was not sourced from cement hydration but from OSP. Therefore, the chemical-bound water content in this test cannot represent the hydration degree of the system. However, the changing trend of compressive strength of each group can be confirmed by the decomposition of C-S-H. As can be seen in Table 6, the introduction of a ternary system containing OSP resulted in a higher CH content. The ternary system containing OSP is different from other types of SCMs. The secondary hydration of LS and GGBFS consumes part of CH, but OSP can supplement this part of CH to maintain the alkalinity balance in the system.

#### 3.2.2. SEM Analyses

Figure 5 shows the morphology of three typical samples at the age of 28 days. It can be found that there were many C-S-H gels in cement paste, and these gels had no obvious defects and have good compactness and continuity. The bond between them seems to be very strong. When the ternary admixture containing OSP was introduced, more CH formed, which is consistent with the TG test results. Figure 5b clearly illustrates the C-S-H gels with two different morphologies. The morphology and density of the C-S-H gel on the left side were similar to those shown in Figure 5a. The C-S-H gel on the right side showed sparse results. This gel was attached to CH more tightly. It can be inferred that the C-S-H gel on the right side in Figure 5b originated from the secondary hydration of the admixture. It is worth mentioning that a similar transition zone existed between the C-S-H gel formed by cement hydration and the secondary hydration of the admixture. This transition zone is a weak zone inside the slurry and has large pores. The presence of this weak area influenced compressive strength, therefore limiting the application of SCMs. Figure 5c shows that a large amount of CH was generated in the system when incorporating OSP as the only admixture, and only a small number of C-S-H gels were attached to the surface of CH. A reduced amount of C-S-H gel caused a decrease in compressive strength of the specimens with the single addition of OSP.

### 3.3. Permeability and Pore Structure Analysis

#### 3.3.1. Water Absorption and Rapid Chloride Penetration Test (RCPT)

Figure 6 shows the water absorption of mortar specimens at the age of 28 days. It was found that the water absorption decreased when increasing the ternary SCMs content, which suggests that the addition of more ternary SCMs in the specimens helped fill capillary pores owing to the second hydration of the SCMs, thus resulting in a lower sorptivity. The single addition of OSP caused a higher permeability compared to the reference group. Additionally, the impermeability of a ternary system was obviously better than that of a binary system or a single OSP addition. In addition to generating more hydration products, OSP, LS, and GGBFS also played a synergistic role in improving particle gradation, which improved the compressive strength and impermeability of the specimens.

Figure 7 illustrates the results of the RCPT of the specimens. It was observed that the samples incorporating the ternary SCM system showed a significant decrease in the total charge passing through, indicating a reduction in porosity. In addition, the single addition of OSP caused a higher charge pass compared to the reference group. The RCPT results influenced transport properties of the specimens through the change of the capillary pore size volume and connectivity, which can be in turn verified via permeability tests. 

#### 3.3.2. MIP Analysis and Pore Structure

Figure 8 compares the MIP curves of samples from the mixtures cured for 28 days. Table 7 shows the porosity parameters of each experimental group. It can be found in Figure 8a that the addition of ternary SCMs resulted in the most significant pore volume reduction and pore size refinement. In contrast, the specimen with the single addition of OSP as SCM showed the largest pore volume, thereby reflecting the lowest compressive strength and highest permeability from the abovementioned sections. Figure 8b shows that the pore structure of the ternary system containing OSP is better than that of single cement when the substitution rates are 10% and 20%. At the hydration age of 28 days, the admixture underwent secondary hydration, and the hydration products increased continuously. The C-S-H gel filled the pores, which reduced the total porosity. At the same time, the admixture particles had a large specific surface area and a good micro-aggregate effect, which filled the pores and reduced the total porosity. When the cement replacement rate increased further to 30%, the secondary hydration degree decreased, and the generated C-S-H gel could not fill more micro-pores, which increased the total porosity. Compared to the pores of the M-1, M-2, M-3, and D-20 groups, the ternary system has the most refined pore structure. In the ternary system, the particle size distribution of the admixture has been optimized, which is more conducive to the formation of a self-compacting system and thus reduces the porosity.

## 4. Conclusions

In this paper, a ternary system concrete admixture containing OSP was proposed, and the compressive strength, permeability, hydration process, hydration products, and porosity were tested. The hydration hardening mechanism and strength variation law of the ternary system containing OSP were revealed. Based on the experimental results, conclusions can be drawn up as follows:(1)There is a coupling effect between OSP, LS, and GGBFS. This coupling effect is based on the self-consistent secondary hydration reaction between calcium hydroxide produced by OSP hydration and LS and GGBFS. The activity index was higher when the substitution rate was 30%.(2)OSP-LS-GGBFS ternary SCMs system has low porosity and permeability was significantly improved. With the increase of SCMs replacement rate, permeability gradually deteriorated.(3)There is a weak transition zone between C-S-H gel produced by secondary hydration and C-S-H gel produced by cement hydration. The gel in this interval is discontinuous and has lots of pores.(4)OSP particles can be activated to provide CH while having good filling effect. Therefore, OSP is a good cement-assisted cementing material, but the single introduction of OSP will cause the problem of too high basicity. Therefore, the multi-cementing system containing OSP will be the direction of future research.

## Figures and Tables

**Figure 1 materials-15-04886-f001:**
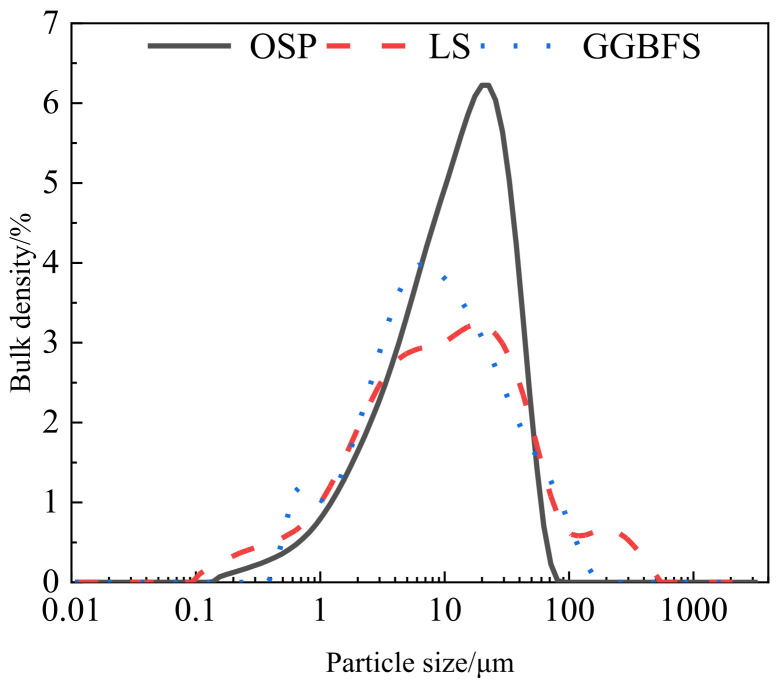
Particle size distribution of materials.

**Figure 2 materials-15-04886-f002:**
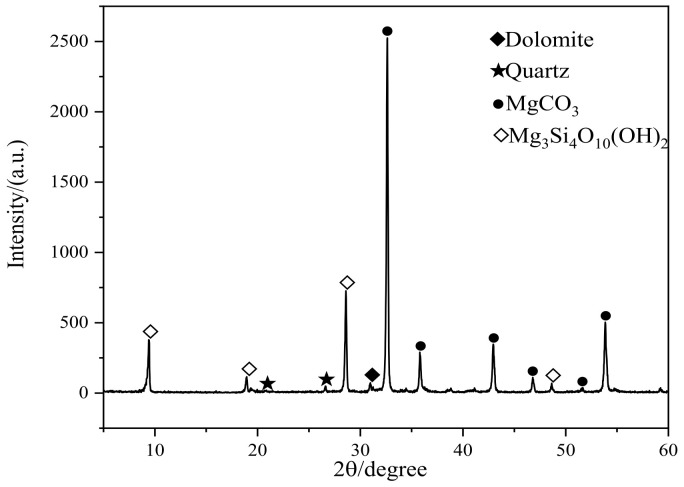
X-ray diffraction patterns of OSP.

**Figure 3 materials-15-04886-f003:**
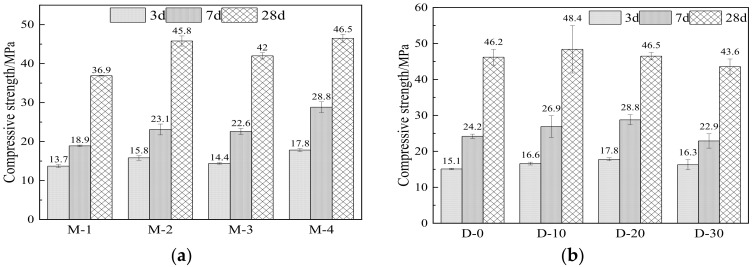
Compressive strength tests. (**a**) The coupling effect of a ternary system, (**b**) replacement ratios of the composite system.

**Figure 4 materials-15-04886-f004:**
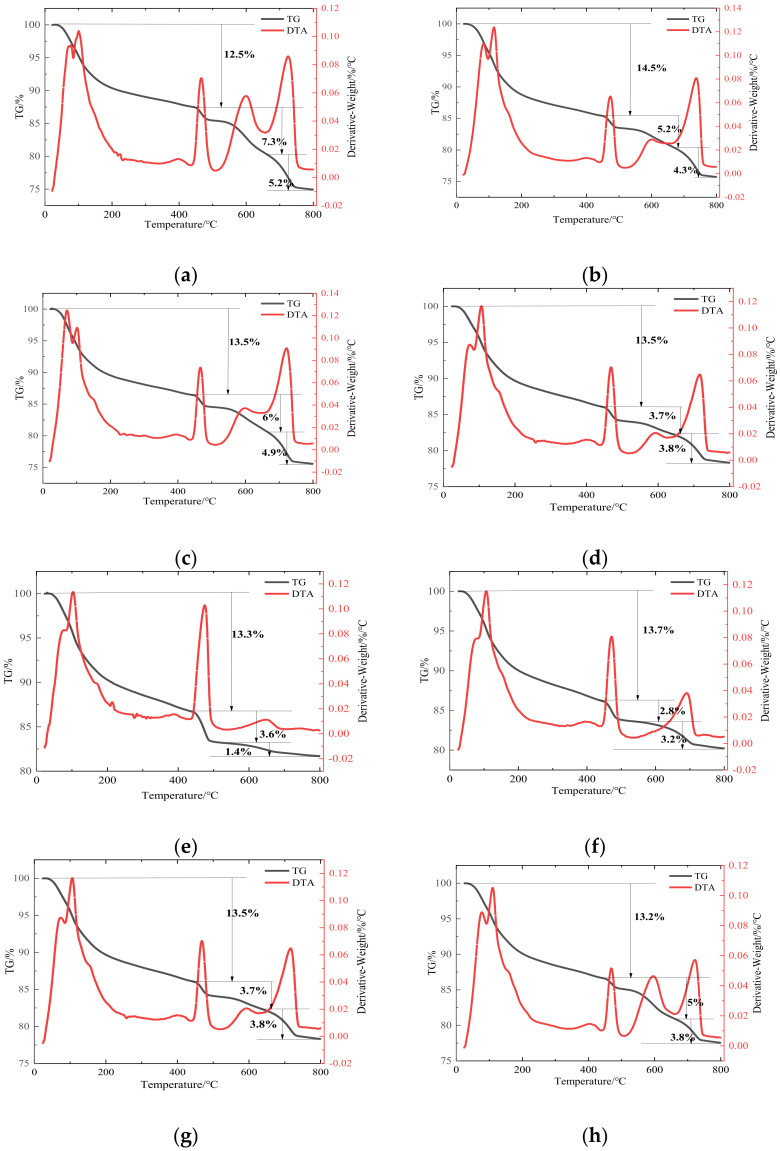
TG test result of each experimental group. (**a**) M−1 group; (**b**) M−2 group; (**c**) M−3 group; (**d**) M−4 group; (**e**) D−0 group; (**f**) D−10 group; (**g**) D−20 group; (**h**) D−30 group.

**Figure 5 materials-15-04886-f005:**
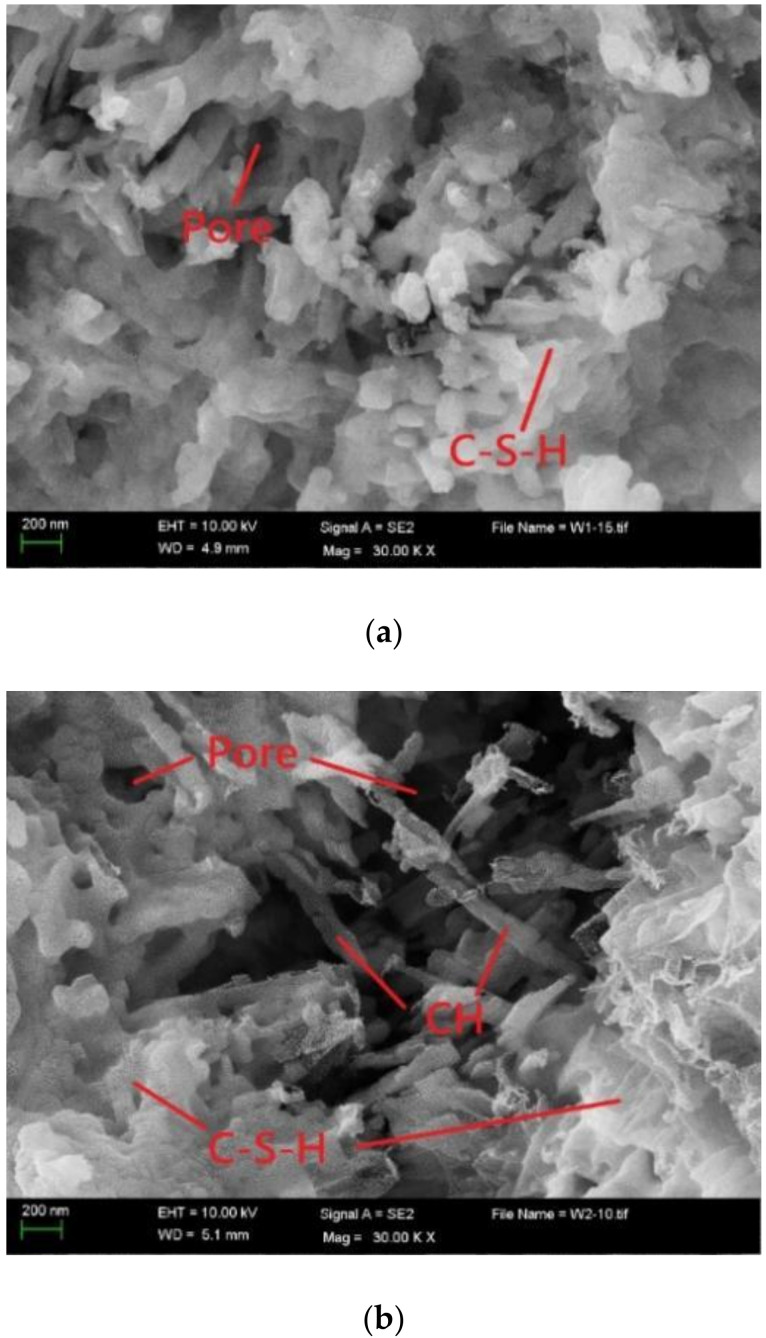
SEM images of hydration products. (**a**) D−0 group; (**b**) D−20 group; (**c**) M−1 group.

**Figure 6 materials-15-04886-f006:**
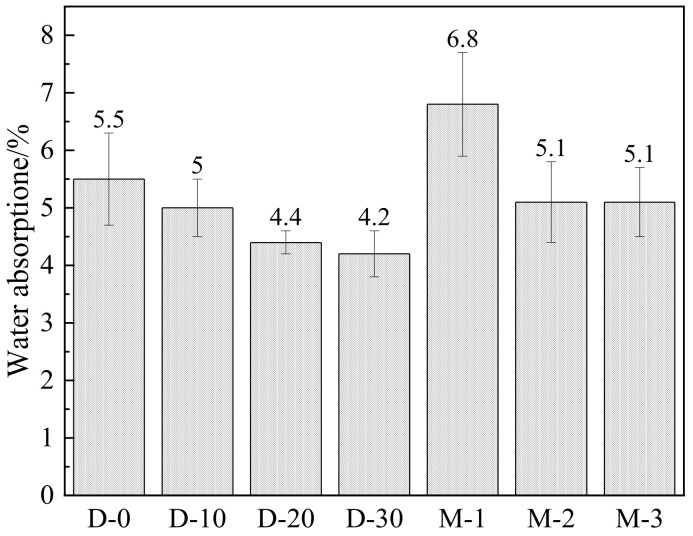
Water absorption in mortars.

**Figure 7 materials-15-04886-f007:**
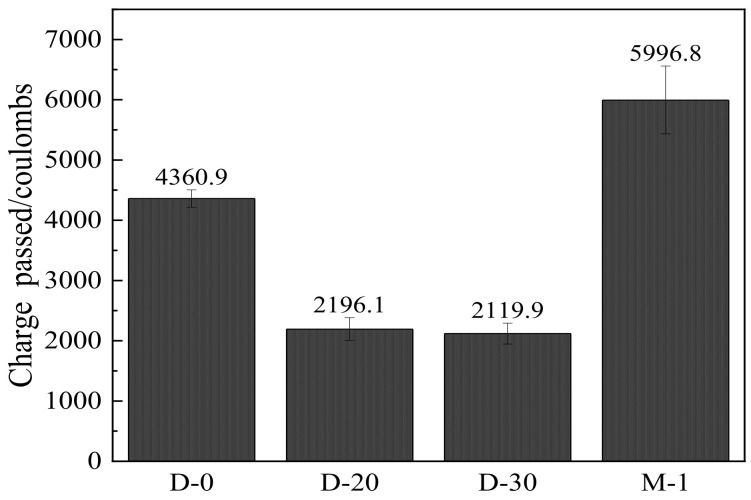
RCPT test results for part of the mortars.

**Figure 8 materials-15-04886-f008:**
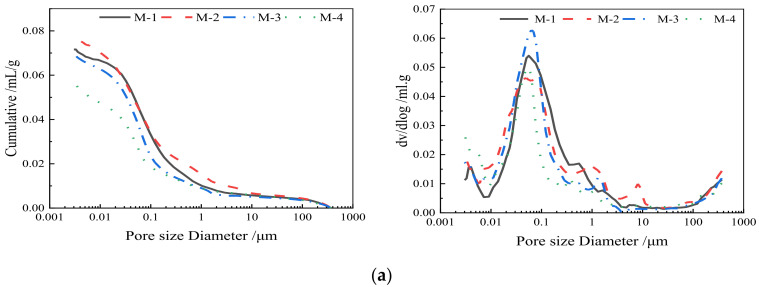
Pore size distributions in mortar samples at 28 days (mL/g). (**a**) Pore distribution curves under different systems; (**b**) Pore distribution curves under different cement replacement ratios.

**Table 1 materials-15-04886-t001:** Chemical composition and content of materials (mass fraction/%).

	SiO_2_	Al_2_O_3_	CaO	SO_3_	MgO	Fe_2_O_3_
OSP	16.5%	0.3%	46.6%	0.05%	36.3%	0.2%
LS	54.5%	25.4%	6.4%	10.2%	0.6%	0.6%
GGBFS	30.7%	15.9%	42.3%	1.8%	6.7%	1.2%
Cement	22.6%	8.3%	61.1%	2.4%	1.9%	2.6%

**Table 2 materials-15-04886-t002:** The specific surface area of the materials.

Materials	OSP	LS	GGBFS
Specific surface/m^2 · ^kg^−1^	2057	13,627	1206

**Table 3 materials-15-04886-t003:** Cement mortar ratio of group M.

Serial Number	SCMs Replacement	Cement/g	OSP/g	LS/g	GGBFS/g	Standard Sand/g	Water/mL
M-1	20%	360	90	0	0	1350	225
M-2	20%	360	45	45	0	1350	225
M-3	20%	360	45	0	45	1350	225
M-4	20%	360	45	22.5	22.5	1350	225

**Table 4 materials-15-04886-t004:** Cement mortar ratio of group D.

Serial Number	SCMs Replacement	Cement/g	OSP/g	LS/g	GGBFS/g	Standard Sand/g	Water/mL
D-0	0	450	0	0	0	1350	225
D-10	10%	405	22.5	11.3	11.3	1350	225
D-20	20%	360	45	22.5	22.5	1350	225
D-30	30%	315	67.5	33.8	33.8	1350	225

**Table 5 materials-15-04886-t005:** Chloride permeability rating.

Chloride Permeability	Charge (Coulombs)
High	>4000
Moderate	2000–4000
Low	1000–2000
Very low	100–1000

**Table 6 materials-15-04886-t006:** TG test the yield table of each substance.

Serial Number	CH to Take Off the Water	Amount of CaCO_3_ Decomposition	C-S-H Decomposition Quantity	H_2_O Content	CH Content
D-0	3.6%	1.4%	13.3%	17.3%	18%
D-10	2.8%	3.2%	13.7%	17.4%	18.8%
D-20	3.7%	3.8%	13.5%	18.5%	23.9%
D-30	5%	3.8%	13.2%	19.5%	29.2%
M-1	7.3%	5.3%	12.5%	21.3%	42.1%
M-2	5.2%	4.3%	14.5%	20.9%	31.2%
M-3	6%	4.9%	13.5%	20.9%	35.8%

**Table 7 materials-15-04886-t007:** Pore characteristic parameter in mortar samples at 28 days.

Serial Number	Total Pore Volume (ml/g)	Maximum Aperture/μm	Pore Size Distribution			
			0–0.02 μm	0.02–0.1 μm	0.1–0.2 μm	>0.2 μm
D-0	0.151	0.095	1.001	0.542	0.073	0.342
D-10	0.139	0.069	0.952	0.518	0.084	0.396
D-20	0.122	0.056	0.747	0.341	0.049	0.301
D-30	0.154	0.051	0.983	0.441	0.063	0.344
M-1	0.148	0.055	1.001	0.546	0.078	0.332
M-2	0.164	0.071	1.071	0.544	0.085	0.428
M-3	0.145	0.062	0.961	0.462	0.055	0.282
M-4	0.122	0.056	0.747	0.341	0.049	0.301

## Data Availability

Not applicable.

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
