# Peer review of "Sustainable Use of Waste Oyster Shell Powders in a Ternary Supplementary Cementitious Material System for Green Concrete"

_materials, 2022, doi:10.3390/ma15144886_

Round 1

Reviewer 1 Report

1) English is serious concern. Please read the paper again and remove the gramatrical error. Also try to write short sentences.

2) Abstract is very weak. Please rewrite it by addressing the methodology and put some results and main findings or conclusions in short comprehensive form in the abstract.

3) In the introduction section, put some literature regarding  oyster shell powders.

4) From where  oyster shell powders was procured. Can you put some pictures regarding that.

5) Pictures and text written on TG analysis is not clear. Kindly make it readable.

6) Do you have EDX analysis along with SEM to confirm your findings?

7) Can you mention, total porosity (MIP) in the tested specimens

8) Add a separate section for the discussion. What are outcomes of this study and compare your results with previous findings.

9) Conclusions are very general. can you be specific and mentioned some increase or decrease in terms of number     

Reviewer 2 Report

The informal language is not suitable and should be improved extensively. The article still needs several grammatical and syntax improvements. Use of English service center is recommended.

·        Majority of the qualitative statements should be modified for quantified result comparisons.  

·        The introduction needs to be revised for higher quality language. The authors mentioned some works without stating about the contributions, pros and cons and the how the current work would address.

·        The purpose of the article should be clarified in details, why this study could be beneficent, more in depth conclusion should be provided.

·        The authors mentioned “have the potential to  replace OPC and aggregates in concrete partially, which not only solves the shortage of  concrete raw materials but also alleviates the environmental pressure caused by waste accumulation ”.  The following references should be added for this statement. 1) Experimental investigation of sound transmission loss in concrete containing recycled rubber crumbs. 2) Nano silica and metakaolin effects on the behavior of concrete containing rubber crumbs. CivilEng. 

·        The application and limitation should be discussed

·        The designed specimens should be elaborated, and design methodology should be referenced.

·        What is the cost analysis between the specimens?

·        Equation used previously should be clearly referenced.

·        From the Fig 3., How the D-30 has 16.3 MPA compressive while D-20 is 17.8 MPA? By increasing from 0 to 10 and then 10 and 30 there was an increase with compressive strength and for 30 it is reduced.

·        How the crack patterns and failure modes are different in the studied specimens?

·        The optimum values and justifications should be determined.

Reviewer 3 Report

The manuscript is well organized and the discussions are clearly stated. The authors need to address the following queries:

Fig 1 "SP" to be replaced with "OSP"

Is the quantity mentioned under Table 3 and 4 have been used for estimating all the properties? Since, the unit is mentioned as ‘g’.

The mortars were placed in molds with a dimension of 40 mm×40 mm×160 mm and vibrated for 120s.” What for this specimen has been prepared?

The test specimen’s size and the corresponding standards needs to be included in section 2.3.

The cylinder sample (U75 mm x 75 mm) was solidified and demoulded” Is the size represents a cylinder?

Why SEM has been done for only 3 samples and M-4 is missing in water absorption and only 4 samples under RCPT? Any specific reason? If so, it has to be included.

Round 2

Reviewer 1 Report

Previous comments addressed.

Reviewer 2 Report

The informal language is not suitable and should be improved extensively. The article still needs several grammatical and syntax improvements. Use of English service center is recommended.